# Cytoprotective, Cytotoxic and Cytostatic Roles of Autophagy in Response to BET Inhibitors

**DOI:** 10.3390/ijms241612669

**Published:** 2023-08-11

**Authors:** Ahmed M. Elshazly, David A. Gewirtz

**Affiliations:** 1Department of Pharmacology and Toxicology, Massey Cancer Center, Virginia Commonwealth University, 401 College St., Richmond, VA 23298, USA; elshazlyam@vcu.edu; 2Department of Pharmacology and Toxicology, Faculty of Pharmacy, Kafrelsheikh University, Kafrelsheikh 33516, Egypt

**Keywords:** autophagy, BET family, BRD4, senescence, cytoprotective, cytotoxic

## Abstract

The bromodomain and extra-terminal domain (BET) family inhibitors are small molecules that target the dysregulated epigenetic readers, BRD2, BRD3, BRD4 and BRDT, at various transcription-related sites, including super-enhancers. BET inhibitors are currently under investigation both in pre-clinical cell culture and tumor-bearing animal models, as well as in clinical trials. However, as is the case with other chemotherapeutic modalities, the development of resistance is likely to constrain the therapeutic benefits of this strategy. One tumor cell survival mechanism that has been studied for decades is autophagy. Although four different functions of autophagy have been identified in the literature (cytoprotective, cytotoxic, cytostatic and non-protective), primarily the cytoprotective and cytotoxic forms appear to function in different experimental models exposed to BET inhibitors (with some evidence for the cytostatic form). This review provides an overview of the cytoprotective, cytotoxic and cytostatic functions of autophagy in response to BET inhibitors in various tumor models. Our aim is to determine whether autophagy targeting or modulation could represent an effective therapeutic strategy to enhance the response to these modalities and also potentially overcome resistance to BET inhibition.

## 1. Introduction

This manuscript is one of a series of papers that highlight the different roles of autophagy in response to various cancer therapeutic modalities. Our previous publications assessed the influence of autophagy in tumor cells on the response/sensitivity to radiation [1], cisplatin [2], microtubule poisons [3], hormonal therapies in estrogen-positive breast cancer [4], PARP inhibitors [5], Topoisomerase I poisons [6] and, most recently, temozolomide [7]. Our overarching goal is to determine whether there are particular therapeutic modalities where the preclinical data, and, where available, clinical trials, support the inclusion of autophagy inhibition as an adjuvant approach.

## 2. Autophagy Overview

Autophagy is an intrinsic cellular process whereby cytoplasmic cargo and damaged organelles are recycled to maintain homeostasis and energy production [3,4]. Autophagy is generally considered as a survival mechanism, with three general types being identified in mammalian cells: microautophagy, macroautophagy and chaperone-mediated autophagy [6]. Macroautophagy, which is the form generally referred to as “autophagy”, is initiated by the formation of double-membraned phagophores. The phagophores elongate and engulf portions of the cytoplasm together with cytoplasmic cargo, such as mitochondria and the endoplasmic reticulum, and then fuse, forming autophagosomes [6]. Subsequently, autophagosomes combine with lysosomes, forming autophagolysosomes, within which the cargo is degraded. A more detailed presentation of the autophagic machinery is provided in our previous publications [3,4]. In addition to the oft-cited cytoprotective function, additional functional forms of autophagy have been established, specifically the cytotoxic, cytostatic and non-protective forms [8]. 

The potential contributions of autophagy to cancer resistance are well established in the scientific literature [9]. The cytoprotective role of autophagy has been explored extensively and multiple clinical trials have been and/or are in progress to determine whether and under which conditions the suppression of autophagy could serve to increase the effectiveness of various chemotherapeutic agents. Chloroquine (CQ) and hydroxychloroquine (HCQ) are the most widely used autophagy inhibitors in pre-clinical studies as well as in clinical trials [3,7]. Recently, several studies highlighting the potential modulation of dysregulated signaling pathways in autophagy have also been published [10,11]. 

In this review, we extensively searched the literature to provide an overview of the relationship(s) between autophagy, the BET family and BET family inhibition in order to evaluate whether autophagy targeting could be a viable strategy to increase the effectiveness of BET inhibitors in different tumor models.

## 3. Super-Enhancers and the BET Family

In eukaryotic cells, protein transcription is initiated by the binding of RNA polymerase to gene promoters. This process is further regulated by the binding of various transcription factors (TFs) to specific DNA sequences in order to recruit factors that mediate RNA polymerase II initiation or elongation. Additionally, DNA sequences located either near or at a distance from promoter regions may contain multiple transcription factor binding sites, referred to as “enhancers” [12]. Enhancers are genomic regions that are bound by TFs and transcriptional co-activators to promote gene transcription [13]. In cancer, large clusters of these enhancers that have been identified are referred to as super-enhancers. Super-enhancers can serve as essential oncogenic drivers required for the maintenance of cancer cell identity [14]. Super-enhancers are co-occupied by various TFs crucial for the relevant cell type and are further inhabited by high levels of transcriptional regulators, including Mediator, p300, CBP, BRD4, RNA polymerase II (RNA Pol II), cohesin and chromatin remodelers [13]. Super-enhancers have recently been identified as potential therapeutic targets in cancer [15,16]. 

The bromodomain is a protein–protein interaction domain, comprising approximately 110 amino acids that recognize and bind acetylated lysine residues in histones, as well as other proteins [17]. Bromodomain-containing proteins (BRDs) serve as epigenetic readers, contributing to the recruitment of transcriptional regulatory complexes to chromatin and binding to acetylated residues in histones [18]. Among the eight families that contain bromodomain modules, the bromodomain and extra-terminal domain (BET) family has attracted extensive attention in recent years [19]. The BET family of proteins is characterized by the presence of two tandem bromodomains and an extra-terminal domain. The mammalian BET family of proteins comprises BRD2, BRD3, BRD4 and BRDT [20], acting as epigenetic readers involved in transcriptional activation, through the recruitment of positive transcription elongation factor (P-TEFb) as well as via the control of RNA-Pol II transcriptional activity [20]. The dysregulated expression of BET family members is involved in many pathological processes and has been recognized as a potential therapeutic target for various diseases, including cardiovascular disease, neurodegenerative disease and cancer [19,21,22,23]. A large number of small-molecule inhibitors targeting the BET family, especially BRD4, have since been developed, including JQ1, the first reported and the most studied BET family inhibitor, which can bind competitively to acetyl-lysine recognition motifs or bromodomains [24], CPI203, MS417 [25] and OTX015 [26]. 

## 4. Autophagy and BRD4

Beyond BRD2 [27] and BRD3 [28] overexpression in different tumor models, BRD4 is the most frequently studied of the BET family members and has been shown to play critical roles in human diseases, including CNS disorders [29], cardiovascular disease [30,31], inflammatory disease [32] and cancer [33,34]. As a component of the general transcription machinery, BRD4 is enriched on hyper-acetylated and transcriptionally prone chromatin regions (both promoters and enhancers), functioning as a nucleation center, recruiting the Mediator complex and promoting the assembly of a large platform of transcription-regulating proteins, forming a bridge between super-enhancers and promoters, favoring and stabilizing the binding of RNA-Pol II. BRD4 also interacts with and activates P-TEFb, stimulating the transition of RNA-Pol II into active elongation (Figure 1). 

Sakamaki et al. [35] investigated the molecular relationship between autophagy and the BET family, focusing on BRD4. Initial studies involving the siRNA-mediated knockdown of genes encoding BRD2, BRD3 and BRD4 in human pancreatic ductal adenocarcinoma KP-4 cells assessed effects on the lipidation of LC3, a protein involved in the development of the autophagosome membrane [4,36]. Knockdown of BRD4, but not BRD2 and BRD3, led to an increase in the levels of LC3II as well as p62/SQSTM1 degradation, both indicative of the promotion of autophagy [37,38,39]. The increase in the levels of LC3II was confirmed by the monitoring of fluorescent LC3 puncta. The use of an RFP-GFP-tandem-tagged LC3 further established that BRD4 knockdown promoted the formation of autophagosomes and subsequent fusion with lysosomes. Conversely, and as would have been expected, the overexpression of BRD4 suppressed autophagic flux in KP-4 cells. Further confirmation that BRD4 acts to repress autophagy was derived from RNA sequencing analysis followed by reverse transcriptase quantitative PCR (RT-qPCR) validation, wherein BRD4 knockdown upregulated the autophagic machinery-related genes that encode proteins involved in autophagosome formation (BECN1, VMP1, PIK3C3, WIPI1, ATG2A, ATG9B and MAP1LC3B) [40], autophagy cargo recruitment (SQSTM1 and OPTN), autophagosome–lysosome fusion (PLEKHM1, TECPR1 and HOPS complex components) [41] and the maintenance of functional ER exit sites and autophagosome formation (MAP1LC3C, TECPR2 and SEC24D) [42]. BRD4 overexpression repressed autophagy gene expression. The general relationship between BRD4 and LC3II was confirmed in different cell lines, including PA-TU-8902, SUIT2, PK-1, PA-TU-8988T, HPNE and HEK293T cells. These results strongly support the premise that BRD4 suppression is associated with autophagy induction. 

## 5. Autophagy and BET Inhibitors

BET inhibitors, an emerging class of agents that target epigenetic dysregulated readers, BRD2, BRD3, BRD4 and BRDT, have recently been studied with a focus on the autophagic machinery. For example, Sakamaki et al. [35] showed that treatment with the largely BET4 inhibitor, JQ1, induced autophagy in KP-4 cells, as evidenced by increased LC3 lipidation and puncta formation. These results were mirrored in vivo using mice models, in which JQ1 treatment caused the conversion of LC3I to LC3II, as well as increasing LC3 puncta formation in mouse small intestinal tissue [35]. The BET inhibitor/degrader, ARV-825, was shown to reduce the levels of BRD2, BRD3 and BRD4 together with inducing autophagy, as shown by increased LC3II levels in KP-4 cells. The BET inhibitors JQ1 and I-BET151, as well as OTX015, upregulated the autophagic genes MAP1LC3B, SQSTM1, ATG2A and ATG16L2, as shown by RT-qPCR [35]. 

In subsequent sections, we discuss the association(s) between autophagy and BET inhibitors in different tumor models.

### 5.1. Bladder Cancer

Li et al. [43] studied the effect of the BET inhibitor JQ1 in the bladder carcinoma cell lines T24, 5637 and UMUC-3. Using an MTT assay, the counting of viable cells and clonogenic survival assays, JQ1 was shown to suppress the proliferation of the three cell lines in a dose-dependent manner. JQ1 treatment increased the number of autophagosomes and autolysosomes based on the GFP-RFP-LC3 fluorescence assay, consistent with autophagy induction [44]. Autophagy induction by JQ1 was confirmed via the accumulation of LC3II and p-ULK1 [45], as well as p62/SQSTM1 degradation and the appearance of autophagic vacuoles by transmission electron microscopy (TEM), all of which was suggestive of a role for autophagy in the anti-proliferative effects of JQ1. To establish whether autophagy was, in fact, mediating drug action in these experimental models, the pharmacologic autophagy inhibitors 3-MA, BAF-A1 and NH_4_Cl were found to interfere with the antitumor activity of JQ1. However, JQ1 could still partially inhibit cellular proliferation, suggesting that mechanistic aspects other than autophagy could also play a role in suppressing the proliferative ability of bladder cancer cells. These findings relating to autophagy were validated by demonstrating that the genetic inhibition of autophagy, using ATG5-directed siRNA, eliminated JQ1′s anti-proliferative actions, consistent with a cytostatic role of autophagy in this model. In this context, a cytostatic function of autophagy was described in early work from our research group [8,46], although there have been few subsequent studies in the literature relating to this component of autophagic action. These results were further mirrored in vivo using a xenograft tumor model implanted with the T24 bladder cancer cell line. Here, JQ1 significantly inhibited tumor growth and induced autophagic flux, as indicated by elevated levels of LC3II and p-ULK1, as well as p62/SQSTM1 degradation.

In additional, quite rigorous mechanistic studies, JQ1 treatment was demonstrated to downregulate p-mTOR (consistent with the promotion of autophagy), accompanied by the upregulation of p-LKB1, p-AMPK*α* and its substrate p-ACC. Using AMPK*α*-directed siRNA, the JQ1 inhibitory effect was eliminated, accompanied by a reduction in autophagy, again shown by monitoring LC3II levels as well as the GFP-RFP-LC3 fluorescence assay. Furthermore, JQ1 could be shown to increase the recruitment of LKB1 and its interaction with AMPK*α* by co-immunoprecipitation assays, suggesting that the autophagy induced by JQ1 is dependent on the LKB1/AMPK/mTOR signaling pathway. 

### 5.2. Ovarian Cancer

Luan et al. [47] investigated the potential contribution of autophagy to JQ1 resistance in ovarian cancer using the A2780, HO-8910, SKOV-3 and HEY cell lines. JQ1 was shown to inhibit the proliferation of the four cell lines, with A2780 and HO-8910 cells showing less sensitivity than SKOV-3 and HEY cells. In these studies, JQ1 was shown to promote concentration-dependent apoptosis (using an annexin V assay). Consistent with the differential sensitivity observed, the SKOV-3 and HEY cells demonstrated a higher apoptosis response as compared to the A2780 and HO-8910 cell lines, in a dose-dependent manner. BRD4 and c-Myc levels were downregulated in the four cell lines upon JQ1 treatment, as would have been expected considering that these are established downstream targets of JQ1 and other BET inhibitors [38,48]. 

Luan et al. [47] further studied whether JQ1 induced autophagy in these preclinical models. In the less sensitive A2780 and HO-8910 cell lines, acridine orange staining, the accumulation of LC3 II, ATG5 and Beclin1 and p62/SQSTM1 degradation were indicative of autophagic flux induced by JQ1 [39]. However, this was not the case with the more sensitive SKOV-3 and HEY cell lines, which showed minimal changes in the autophagic markers with JQ1. These results suggested that JQ1-induced autophagy may have contributed to the reduced sensitivity to JQ1 in the A2780 and HO-8910 cells. The combination of JQ1 with the pharmacologic autophagy inhibitors, 3-MA and CQ, sensitized the A2780 and HO-8910 cell lines and increased the extent of apoptosis, as shown by annexin V staining and cleaved-PARP levels. Taken together, this series of experiments suggested a cytoprotective role of autophagy in two of the four cell lines. This cytoprotective role of autophagy was further confirmed in vivo using xenograft tumor models implanted subcutaneously with A2780 cells; specifically, JQ1 in combination with CQ showed higher antitumor activity than each drug alone in the mice models.

Luan et al. [47] further studied the relation between JQ1-induced autophagy and the Akt/mTOR pathway. JQ1 treatment in the less sensitive A2780 and HO-8910 cell lines caused a reduction in the phosphorylation levels of Akt, and, consistent with the findings of Li et al. [43], mTOR or p70S6K. However, this was not the case in the JQ1-sensitive SKOV-3 and HEY cell lines, where JQ1 did not affect the levels of Akt, mTOR and p70S6K. Overall, these experiments are consistent with a cytoprotective function of autophagy in preclinical models of ovarian cancer.

### 5.3. Breast Cancer

Ali et al. [49] studied the possible targeting of the BRD4/RAC 1 axis using the MCF-7, MDA-MB-231, JIMT1 and SKBR3 breast tumor cells lines, also using JQ1. BRD4 inhibition, using JQ1 in combination with the RAC1 inhibitor NSC23766, suppressed cellular growth, clonogenic survival, cell migration and mammary stem cell expansion. The anti-proliferative effect of the combination was confirmed in vivo, using the MDA-MB-231-based xenograft model, where JQ1 in combination with NSC23766 showed significant combined anti-proliferative effects. Here, JQ1 in combination with NSC23766 induced senescence in MCF-7, MDA-MB-231 and JIMT1 cells, but not in SKBR3 cells. The combination treatment resulted in the greater accumulation of LC3-II than each drug alone, suggesting that the combination of JQ1 and NSC23766 may induce (either cytostatic or cytotoxic) autophagy. However, these studies did not further explore the potential involvement of autophagy in JQ1 action, in contrast to the studies described above in the ovarian cancer and bladder cancer cell work. 

### 5.4. Glioblastoma

Colardo et al. [50] studied the relationship between the BET family and autophagy in glioblastoma (GBM) using U87MG (U87), GL15 and the patient-derived GH2 cell lines. These investigators reported that the expression levels of BRD2 and BRD4 were high in these three cell lines, as well as in GBM patient samples. JQ1 significantly suppressed the proliferation of both the U87 and GH2 cell lines. Furthermore, upon combining JQ1 with temozolomide, the standard-of-care of therapy for GBM [7], a greater anti-proliferative response was evident compared to each drug alone, together with increased apoptosis. JQ1 treatment also promoted marked morphological changes in both cell lines, such as increasing the number of GBM cells with cytoplasmic extensions and elongation, suggesting the induction of a differentiation process. The induction of differentiation was further confirmed by the accumulation of synaptophysin and β3-tubulin, early markers of neuronal differentiation [51,52]. 

With respect to autophagy, the protein levels of autophagy-related markers ULK1, ATG5, ATG7, Beclin1 and p62/SQSTM1 were examined by Western blotting in U87 and GH2 cells [50]. Here, JQ1 caused the transient upregulation of ULK1, an upstream promoter of autophagy, but without promoting significant differences in ATG5, ATG7 and Beclin1 protein levels. Furthermore, a transient elevation in p62/SQSTM1 levels at 24 h after JQ1 treatment was observed, followed by a significant reduction, which was not prevented by CQ, suggesting that there was an impairment in p62/SQSTM1 expression instead of p62/SQSTM1 degradation. Despite these apparent inconsistencies, JQ1 induction of autophagy was confirmed in U87 and GH2 cells by immunofluorescence analysis of endogenous LC3 with the accumulation of LC3 dots [50]. 

The relationship between GBM cell differentiation and autophagy was investigated using Beclin1-directed shRNA in GL15 cells [50]. Upon combining genetic autophagy inhibition with JQ1, β3-tubulin and synaptophysin levels were lower than in the shRNA control samples. These results were confirmed with the pharmacological inhibition of autophagy using CQ, where the JQ1-induced accumulation of β3-tubulin and synaptophysin was suppressed upon CQ treatment, confirming the importance of the autophagic flux in JQ1-induced GBM differentiation [50]. 

The role of BET family members in regulating stem cell differentiation, through the involvement of various signal transduction pathways, has been studied previously [53,54]. BRD4 was shown to be occupied at the Notch1 promoter site, thus controlling the Notch1 signaling pathway that is involved in regulating the self-renewal capacity of glioma stem cells and their tumorigenicity [55]. Interestingly, Li et al. [56] reported that BET inhibition caused an increase in the number of neurons. Furthermore, gene expression profiling analysis demonstrated that BET bromodomain inhibition induced a transcriptional program enhancing the directed differentiation of neural progenitor cells into neurons, while suppressing cell cycle progression and gliogenesis. Regarding autophagy, it is well known that autophagy plays an important role during embryonic development and differentiation, maintaining cellular homeostasis as well as the stemness characteristics of self-renewing cells [57,58,59]. Therefore, Colardo et al. [50] proposed that dysregulated BET protein expression negatively regulates autophagy in GBM cells, maintaining stemness and contributing to tumor aggressiveness. 

These data raise the question of whether a form of autophagy, either cytoprotective, cytotoxic or cytostatic, plays a prominent role in the differentiation process. It would appear to be the cytostatic form of autophagy, consistent with growth arrest in differentiated cells. Furthermore, in this study, JQ1 suppressed cell proliferation in the glioblastoma cells. However, additional experimental data would be needed to confirm the proposed cytostatic nature of autophagy, since Colardo et al. [50] did not investigate the role of the autophagic flux in this model.

### 5.5. Pancreatic Cancer

Xu et al. [60] studied the combination of JQ1 and arsenic trioxide (ATO) in pancreatic cancer. Using 11 pancreatic cancer cell lines, these investigators observed that ATO effectively induced ER stress, an unfolded protein response (UPR) [7] and eventually apoptosis in some cell lines, including B×PC-3 and MIAPaCa-2 cells, independently of K-ras or p53 status. Furthermore, upon analyzing ATO-treated B×PC-3 microarray data, autophagy-related genes were upregulated, including GABARAPL1, GABARAPL2, ULK1 and ATG12, which was further validated using RT-PCR analysis. Autophagy activation was further confirmed by Western blotting, indicating the upregulation of LC3-II, ATG5, ATG7 and Beclin1 protein levels; however, p62/SQSTM1 protein as well as mRNA levels were elevated, which would appear to be a contradictory outcome indicative of autophagy inhibition. Therefore, using a GFP assay as well as microarray data, these authors showed that ATO promoted immature autophagosome accumulation via activating autophagosome formation but also inhibiting the lysosomal functions and consequently degradation. Treatment with the autophagy inhibitor CQ further reduced the viability of B×PC-3 and MIAPaCa-2 cells, suggesting that a cytotoxic form of autophagy resulted from the accumulation of autophagosomes and impaired lysosomal function, leading to cell death [7]. Autophagy induction was also shown to result from ATO-induced ER stress, in which TEFB and TFE3 may play important roles. 

Nuclear factor (erythroid-derived 2)-like 2 (NRF2) is a basic leucine zipper transcription factor within the cap “n” collar family [61]. NRF2 regulates the activity of many enzymes and transporters that are involved in fatty acid synthesis and oxidation, xenobiotic detoxification and transportation, as well as conjugation reactions [62,63]. Furthermore, NRF2 regulates the activity of other transcription factors, including AhR, PPAR γ, CEBPα and RXRα [63,64,65]. Recently, attention has been directed to the possible relationships between NRF2 and cancer [66]. Wu et al. [63] suggested that NRF2 is a double-edged sword. NRF2 signaling pathways appear to be responsible for protection against chemical-induced oxidative damage, maintaining redox homeostasis and exerting anti-inflammatory as well as antineoplastic activity. However, NRF2’s persistent activation has been associated with metabolic reprogramming, apoptosis suppression and increasing the self-renewal abilities of cancer stem cells, as well as chemotherapeutic resistance [63]. 

Interestingly, Xu et al. [60] showed that ATO treatment altered the NRF2 expression profile, causing a decrease in the sensitive cells and an increase in the insensitive cells. Moreover, upon NRF2 knockdown, ATO treatment resulted in reduced cell viability and the induction of an ER stress response, as well as apoptosis in insensitive cells. Similar results were generated in mice models implanted with NRF2 knockdown cells, in which ATO treatment resulted in a significant reduction in tumor size [60]. To investigate whether a possible relationship exists between autophagy and NRF2 in pancreatic cancer, NRF2 was depleted in the ATO-insensitive PANC-1 cell line. Here, autophagic genes were significantly elevated in the NRF2 knockdown cells upon ATO treatment. Autophagy induction was confirmed by the accumulation of LC3II, as well as by TEM-detected autophagosome formation [60]. Moreover, lysosomal-related genes/proteins were suppressed in the NRF2 knockdown cells upon ATO treatment, suggesting the impairment of lysosome-related activity.

JQ1 sensitized the ATO-resistant cell lines, including PANC-1, YAPC, AsPC-1, HAPF-II, CFPAC-1 and HUP-T4 cells. JQ1 treatment led to the significant downregulation of NRF2 levels, as assessed by an immunoblotting assay, without affecting NRF2 mRNA, suggesting that NRF2 is regulated by BET proteins via indirect mechanisms, including translation or protein stability control [60]. Furthermore, the combination of JQ1 and ATO produced a greater reduction in tumor size than each drug alone in mice models implanted with pancreatic cancer cells. JQ1 in combination with ATO generated a further increase in the levels of LC3 II, ATG-5, ATG-7 and Beclin1, together with p62/SQSTM1 accumulation, than each drug alone [60]. Interestingly, these findings were accompanied by a reduction in the lysosomal protein CTSB, suggesting the impairment of lysosome function. The impairment of lysosomal function was further confirmed using immunofluorescent assays. Upon combining ATO and JQ1 together with CQ, the viability of the ATO/JQ1-treated population was significantly reduced, suggesting a possible cytotoxic role of autophagy due to the accumulation of autophagosomes, leading to cell death [7]. Therefore, they proposed that JQ1 exerts an effect similar to NRF2 knockdown in sensitizing pancreatic cancer cells to ATO-induced autophagosome accumulation, ER stress/UPR and apoptosis [60].

### 5.6. Leukemia

Jang et al. [67] studied the relationship between autophagy and JQ1 resistance in leukemia stem cells (LSC), using the KG1, KG1a and Kasumi-1 cell lines, which are enriched in the LSC phenotype. JQ1 inhibited the proliferation of the Kasumi-1 cell line, with a lesser anti-proliferative response noted with KG1 and KG1a cells. Moreover, the apoptotic response to JQ1 varied, with the highest apoptotic population evident in Kasumi-1 cells, with minimal apoptosis in the KG1 and KG1a cell lines. These results were mirrored in AML patient samples, with variability in the apoptotic response to JQ1. 

Mechanistically, JQ1 was able to suppress c-Myc expression in both JQ1-sensitive and insensitive cell lines [67]. JQ1 induced autophagy in the KG1 and KG1a cell lines but not in Kasumi-1 cells, as evidenced by the conversion of LC3I to LC3II, as well as by p62/SQSTM1 degradation. The autophagy induction was further confirmed using TEM as well as GFP-LC3 puncta, showing a significant increase in the number of LC3 puncta in the KG1 and KG1a cell lines but not in Kasumi-1 cells [67]. Autophagy induction was also confirmed in JQ1-resistant patient samples, as shown by increased LC3I/LC3II conversion and an increase in the number of GFP-LC3 puncta, suggesting a possible relationship between autophagy and JQ1 resistance, i.e., that autophagy was demonstrating a cytoprotective function. This assumption was confirmed with pharmacologic autophagy inhibition using bafilomycin A1, 3-MA or hydroxychloroquine, and genetically using Beclin1-directed siRNA [67]. Autophagy inhibition sensitized the resistant cell lines and patient samples to JQ1, as confirmed by increased apoptosis and cleaved caspase-3 and cleaved-PARP accumulation, suggesting a cytoprotective role of autophagy in the resistant cells.

On the molecular level, Beclin1 levels were observed to be increased in parallel with LC3II in the resistant cell lines, which was not the case in the JQ1-sensitive Kasumi-1 cells [67]. Furthermore, JQ1 increased the phosphorylation of AMPK (Thr172), ULK1 (Ser555), mTOR (Ser2448) and p70S6K in the resistant cells (in contrast to both Li et al. [43] and Luan et al. [47], confirming that autophagy is tumor/cell type-specific).

Wang et al. [68] reported that the overexpression of c-Myc by tumor cells is needed to maximize glycolysis and oxidative phosphorylation in order to support the high level of ATP consumption required by rapid, proliferation-associated anabolism in tumor cells [69,70]. Therefore, c-Myc inhibition is accompanied by metabolic de-regulation, mitochondrial atrophy, neutral lipid accumulation, cell cycle arrest, ATP depletion and an effort to replenish ATP via the upregulation of AMPK. Therefore, Jang et al. [67] proposed that JQ1 suppresses c-Myc levels, which, in turn, depletes ATP stores and induces the phosphorylation of AMPK [68]. The AMPK-mediated phosphorylation of ULK1 induces autophagy, which can be responsible for JQ1 resistance, independently of the mTOR pathway. The role of AMPK in JQ1 resistance was confirmed via AMPKα-targeted siRNA, as well as by the use of the AMPK pharmacological inhibitor, compound C. AMPK inhibition increased the extent of the apoptotic response in JQ1-treated resistant cell lines and patient samples, consistent with autophagy expressing a cytoprotective function. 

On the other hand, JQ1 increased AMPK phosphorylation without p-ULK1 accumulation or autophagy induction, in Kasumi-1 cells and JQ1-sensitive patient samples, with a reduction in both mTOR and p70S6K phosphorylation. It was therefore proposed that the failure of JQ1 to promote ULK1 phosphorylation was the basis for the absence of autophagy induction. However, JQ1 increased the levels of apoptotic protein markers, including cleaved caspase-3, cleaved caspase-9 and PARP. These data raise the question as to why ULK-1 is induced in the JQ1-resistant cell line, leading to the induction of cytoprotective autophagy, and not the JQ1-sensitive cell lines, despite AMPK activation and mTOR inhibition. 

### 5.7. The Contribution of Autophagy to Senolysis Mediated by BET Inhibition

Senescence is defined as a biological state in which cells have lost the ability to divide, but remain metabolically active for a defined period of time [71,72,73,74]. Three types of senescence have been identified in the literature: replicative senescence, oncogene-induced senescence and premature (accelerated, therapy- or stress-induced) senescence [75,76]. Senescence was recently identified as a hallmark of cancer [77]; furthermore, recovery from senescence could contribute to disease recurrence and is often associated with tumor aggressiveness and therapy resistance [71,78,79]. Senolytics, senostatics and senomorphics have been investigated recently for the possible elimination or at least extended suppression of dormant tumor cell populations [79,80,81,82]. Among different types of senescence-targeted drugs, senolytics recently emerged as a potential adjuvant therapy in combination with various chemotherapeutic modalities. Senolytics eliminate the senescent cells via targeting critical proteins involved in pro-survival and anti-apoptotic mechanisms, such as p53, p21 and Bcl-2 family proteins [83]. Our laboratory, as well as other researchers [84,85], has investigated the utilization of various senolytics, including ABT-263, as a possible strategy for the elimination of the senescent cells that are induced by etoposide, doxorubicin, cisplatin and radiation by interfering with the interaction between Bcl-xl and BAX in different tumor models [81,86]. 

Recently, Wakita et al. [87] studied the potential utilization of BET inhibitors/degraders as senolytics. Here, we are moving away from JQ1, which cannot be considered for clinical utilization due to unacceptably high levels of toxicity [88,89,90]. ARV-825 was shown to have potential senolytic activity using different senescence models, specifically human diploid fibroblasts (HDFs) driven into a senescent state via serial passage (replicative senescence), treatment with doxorubicin (therapy-induced senescence) and infection with a retrovirus encoding oncogenic Ras (+HRasV12) (oncogene-induced senescence). In these studies, ARV-825 eliminated senescent cells in a dose-dependent manner, together with a reduction in both BRD3 and BRD4 protein levels. To further confirm these findings, the senolytic activity of ARV-825 was examined in vivo using an obesity-induced hepatocellular carcinoma (HCC) mouse model. Here, elevated activity of the gut bacterial metabolite deoxycholic acid (DCA) caused DNA damage and provoked cellular senescence and SASP production in hepatic stellate cells (HSCs), which in turn led to HCC development in neighboring hepatocytes through SASP [91]. ARV-825 treatment resulted in a significant reduction in HCC development together with a reduction in senescent HSC numbers, as validated by the immunofluorescence (IF) staining analysis for αSMA (activated HSC marker) and p21 (senescence marker) or IL6/Groα (SASP marker). These results were further validated in vitro using cultured senescent HSCs provoked by DCA treatment, where ARV-825 showed marked senolytic activity. In another model of therapy-induced senescence, induced by treating HCT116 colon cancer cells with doxorubicin, ARV-825 preferentially eliminated the senescent population. Furthermore, treating HCT116 xenograft mice models with ARV-825 in combination with doxorubicin was more effective than the use of doxorubicin alone. This reduction in the tumor size with the combination was accompanied by a reduction in the senescence markers, p21Waf1^/Cip1/Sdi1^ expression and 53BP1 foci formation.

Mechanistically, it was shown that ARV-825’s senolytic activity is dependent on BRD4 as siRNA-based depletion of BRD4; neither BRD2 nor BRD3 depletion markedly promoted senolysis in HDFs. However, unexpectedly, RNA sequencing analysis indicated that ARV-825 failed to downregulate c-Myc, Bcl-Xl and Bcl-6, the downstream targets of BRD4 [87,92,93]. Interestingly, ARV-825 was shown to inhibit the non-homologous end joining (NHEJ) repair for DNA double-strand breaks (DSBs) in the senescent population; concomitantly, RNA sequencing indicated that ARV-825 treatment elevated the expression of the autophagic genes MAP1LC3B, p62/SQSTM1 and ATGA2, together with increasing LC3 puncta in senescent HDFs. Critically, the treatment of senescent HDFs with autophagy inhibitors, bafilomycin A1 or chloroquine, substantially decreased the ARV825-induced senolysis, suggesting that autophagy may have a cytotoxic role in the senescent population.

In studies of HCT116 xenografts, ARV-825 treatment in combination with doxorubicin resulted in an increase in cleaved caspase-3 and γH2AX (a DNA damage marker), as well as LC3II, in tumor tissue. Furthermore, the combined use of CQ with ARV825 largely eliminated the tumor-suppressive effect of ARV825 in HCT116 xenograft mice models treated with doxorubicin. Therefore, Wakita et al. [87] proposed that BRD4 inhibition via ARV-825 provokes senolysis by activating autophagic flux, through DSB exacerbation as well as the upregulation of autophagic genes in senescent cells (i.e., a cytotoxic function of autophagy).

These results are, in part, consistent with data from our laboratory, where we investigated the potential utilization of ARV-825 as a senolytic in ER+ breast cancer models [38]. Initially, senescence was induced in both MCF-7 and T47D cells using a combination of fulvestrant plus palbociclib (F+P), as confirmed by beta-galactosidase staining and the quantification of senescence using C_12_ FDG, a beta-galactosidase florescent metabolite, as well as by monitoring SASP markers. ARV-825 caused a reduction in the senescent population and promoted apoptosis. We further confirmed the reduction in the extent of senescence via C_12_ FDG. Consistent with the findings of Wakita et al. [87], we showed that ARV-825 reduced BRD4 levels in both the MCF-7 and T47D cell lines. However, in contrast to the studies by Wakita et al. [87], ARV-825 reduced c-Myc levels in both cell lines. Interestingly, we found that ARV-825 induced autophagy in the F+P-treated population compared to the controls, as shown by acridine orange staining (unpublished data), suggesting a possible relation between autophagy and senolysis induced by ARV-825.

## 6. Conclusions

There has been growing interest in the potential targeting of dysregulated epigenetic regulators, such as “super-enhancers”, in cancer treatment [94]. One of the transcription factor families that occupy these enhancer sites is the BET family. The BET family comprises different members, including BRD2, BRD3, BRD4 and BRDT. The relationship between autophagy and the BET family has been studied in a number of publications. Autophagy induction has been shown to be associated with various therapeutic modalities, with different functions, including cytotoxic, cytoprotective, cytostatic and non-protective forms. Importantly, and as mentioned in previous publications [3,4,5,6,7] and largely accepted in the literature, the nature of the induced autophagy is dependent on the cell line as well as the nature of the chemical compound being utilized. As summarized in Table 1, the majority of the studies showed that BET inhibitors induced autophagy in different tumor models with two main roles, cytotoxic and cytoprotective. However, cytostatic autophagy was also identified in two studies, whereas no evidence for non-protective autophagy has yet been shown for BET inhibitors. 

The association between BET inhibitor resistance and autophagy induction reported by Luan et al. [47] in ovarian carcinoma and Jang et al. [67] in leukemia suggests that autophagy may play a role in the development of resistance and, consequently, that BET targeting may represent a productive clinical strategy. Studies by Colardo et al. [50] also showed a possible association between autophagy induced by BET inhibition and neuronal differentiation, which may reflect the cytostatic form of autophagy, as in the case of the studies by Li et al. [43]. Furthermore, it is notable that studies both from our laboratory [38] as well as by Wakita et al. [87] have implicated autophagy as contributing to the elimination of the senescent population by ARV-825. 

Nevertheless, the relationships between BET inhibitors and autophagy are still unclear and will require further investigation, since three of the four known forms of autophagy have been identified in the published literature. Furthermore, additional autophagic markers and assays, and particularly genetic strategies, will be required to rigorously establish the role of autophagy induced in each tumor model, as emphasized in the guidelines established by Klionsky et al. [95].

## Figures and Tables

**Figure 1 ijms-24-12669-f001:**
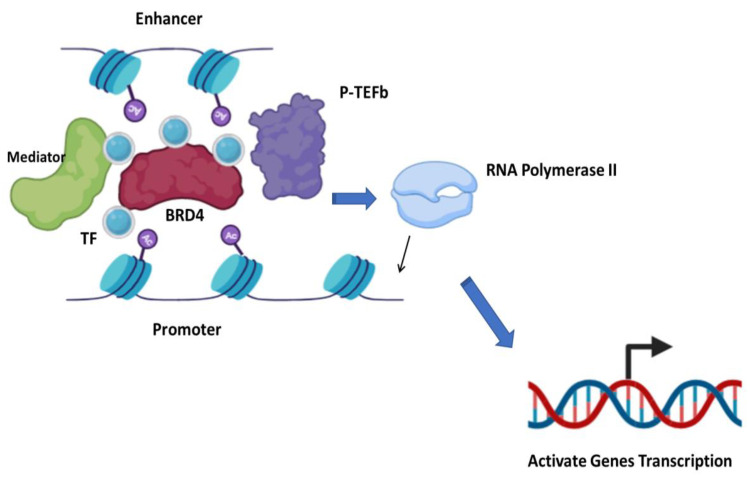
BRD4 regulation of gene transcription. BRD4 binds promoter and enhancers, forming a bridge by binding to various transcription factors and Mediator complex, supporting the binding of RNA polymerase II for transcription. BRD4 also activates P-TEFb, stimulating transition of RNA-Pol II into active elongation.

**Table 1 ijms-24-12669-t001:** BET inhibitors and the roles of autophagic machinery induced in different tumor models.

BET Inhibitor	Cell Lines/Tumor Type	Autophagy Induced/Suppressed	Role of Autophagy	References
JQ1, ARV-825, I-BET151 and OTX015	human pancreatic ductal adenocarcinoma KP-4 cells, and in vivo using mice models	Autophagy induced	NA	[35]
JQ1	T24, 5637 and UMUC-3 bladder cancer cell lines, and in vivo using xenograft tumor model injected with T24 cell line	Autophagy induction	Cytostatic	[43]
JQ1	A2780, HO-8910, SKOV-3 and HEY ovarian cancer cells, and in vivo using xenograft tumor models implanted subcutaneously with A2780 cells	Autophagy induction in A2780 and HO-8910 cells but not in SKOV-3 and HEY cells	Cytoprotective in A2780 and HO-8910 cells	[47]
JQ1	MCF-7, MDA-MB-231, JIMT1 and SKBR3 breast cancer cells, and in vivo using MDA-MB-231-based xenograft tumor mice	JQ1 and NSC23766 induced autophagy	Either cytostatic or cytotoxic	[49]
JQ1	U87MG (U87), GL15, the patient-derived GH2 GBM cell lines and GBM patient samples	Autophagy induced	Autophagic flux has a role in JQ1-induced GBM differentiation	[50]
JQ1	Using 11 pancreatic cancer cell lines, including B×PC-3, MIAPaCa-2, PANC-1, AsPC-1, YAPC, CFPAC-1, HAPF-II and HUP-T4 cells. In vivo using mice models injected with pancreatic cancer cells.	JQ1 in combination with ATO induced autophagy	Cytotoxic	[60]
JQ1	KG1, KG1a and Kasumi-1 leukemia cell lines as well as AML patient samples	Autophagy induced in KG1 and KG1a but not in Kasumi-1	Cytoprotective	[67]
ARV-825	Using different senescence models, human diploid fibroblasts (HDFs) driven into a senenscence state via serial passage (replicative senescence), treatment with doxorubicin (therapy-induced senescence) as well as infection with retrovirus encoding oncogenic Ras (+HRasV12) (oncogene-induced senescence) HCT116 colon cancer cells HCT116 xenograft mice Hepatic stellate cells treated with DCA Obesity-induced hepatocellular carcinoma mouse model	Autophagy induced in the senescence population	Cytotoxic in the senescence population	[87]
ARV-825	MCF-7 and T47D ER+ breast cancer cell lines	Autophagy induced	Cytotoxic in senescent population	[38]

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
