# Peer review of "Cytoprotective, Cytotoxic and Cytostatic Roles of Autophagy in Response to BET Inhibitors"

_ijms, 2023, doi:10.3390/ijms241612669_

Round 1

Reviewer 1 Report

1. BET inhibitors and autophagy are central point of this review article. However author didn't mentioned about BET in abstract as well as in introduction not even full abbreviations until line no 82. Author may add few sentences about BET in abstract as well as in introduction.

2. In line no. 44 and 45 same reference cited. Author may use once instead of twice.

3. in line no 89 author mentioned " BET family members is involved in many pathological processes" author may add 1-2 examples.

4. Headings are not uniform. Author used number up-to 4 only. Rest of the headings are without number. 

5. In "Autophagy and BET inhibitors" author didn't cite any reference with used text. 

6. In glioblastoma second and third paragraph,  Author didn't cite any references. Author may add relevant citations. 

7. In pancreatic cancer third and fourth paragraph author may use relevant citation with the text.

8. In leukemia second paragraph author may add relevant citation. 

9. Author may add a paragraph to discuss the role of cytoprotective, cytotoxic and cytostatic mechanism used in the body to maintain homeostasis and which type of pathological changes appears when cytoprotective changed to cytotoxic or cytostatic or vice-versa. 

Reviewer 2 Report

This manuscript is an extension of previous work reviewed by Dr. Gewirtz and collaborators on the role of autophagy in cancer therapy. It is well written and would provide a useful addition to the literature. The following comments and suggestions are not major in nature and are submitted for consideration by the authors.

Lines 108-117 and lines 132-138 have some overlap in content and might be shortened by reference back to the first passage in the second passage.

Line 249, the last word could be expanded from shcontrol to shRNA control.

Line 53.54 and line 289, CQ is introduced as an autophagy inhibitor and is generally referred to as such except on line 289 where it is described as a lysosomal inhibitor. Is there a reason for that discrepancy?

Line 294, Insert the word “zipper” between “leucine” and “transcription”.

Line 403, Insert the word “of” between “elimination” and “the”.

Line 480. Luan et al. are noted as working on bladder carcinoma but references 45 presents work on ovarian cancer.

Line 511, the authors of reference 5 should be presented using lower case letters to be consistent with the other references.
